# Olive Oil as a Transport Medium for Bioactive Molecules of Plants?—An In Situ Study

**DOI:** 10.3390/molecules28093803

**Published:** 2023-04-28

**Authors:** Jasmin Flemming, Clara Theres Meyer-Probst, Kristin Hille, Sabine Basche, Karl Speer, Isabelle Kölling-Speer, Christian Hannig, Matthias Hannig

**Affiliations:** 1Clinic of Operative Dentistry, Medical Faculty Carl Gustav Carus, Technische Universität Dresden, Fetscherstraße 74, D-01307 Dresden, Germany; 2Special Food Chemistry and Food Production, Technische Universität Dresden, Bergstraße 66, D-01069 Dresden, Germany; 3Clinic of Operative Dentistry, Periodontology and Preventive Dentistry, University Hospital Saarland University, Building 73, D-66421 Homburg, Germany

**Keywords:** initial bacterial colonization, erosion, in situ, polyphenols, ellagic acid, olive oil

## Abstract

(1) Caries and erosions still remain a challenge for preventive dentistry. Certain plant extracts have shown beneficial effects in preventive dentistry. The aim of this study was to evaluate the antibacterial, anti-adherent and erosion-protective properties of ellagic acid (EA) as a polyphenolic agent. The combination with olive oil was investigated additionally to verify a possible improved bioactive effect of EA. (2) An in situ study was carried out with six subjects. Individual splints were prepared with bovine enamel specimens. The splints were worn for 1 min (pellicle formation time). Thereafter, 10 min rinses were performed with EA in water/in oil. Bacterial adherence was evaluated by fluorescence microscopy (DAPI, ConA, BacLight) after an 8 h oral exposition time. Additionally, the splints were worn for 30 min to quantify demineralization processes. The ultrastructure of the pellicle was investigated after an oral exposure time of 2 h under a transmission electron microscope. Statistical analysis was performed by Kruskal–Wallis tests, Mann–Whitney U tests and Bonferroni–Holm correction. (3) Rinsing with EA led to a significant reduction of adherent vital and dead bacteria. The combination with olive oil did not improve these outcomes. The assessment of glucan structures after rinsing with EA in water showed significant effects. Significant differences were observed for both rinses in calcium release at pH 3.0. After rinsing with EA in oil, significantly less calcium was released compared to rinsing with EA in water (pH = 3.0). (4) Olive oil is not suitable as a transport medium for lipophilic polyphenols. EA has anti-adherent and antibacterial properties in situ. EA also shows erosion-protective effects, which can be enhanced in combination with olive oil depending on the pH value. Ellagic acid has a neutral pH and could be an opportunity in the treatment of specific patient groups (xerostomia or mucositis).

## 1. Introduction

Global caries prevalence has decreased considerably within the last 50 years [1]. This is mainly attributed to the widespread use of fluorides in private and professional applications [1,2,3]. However, sugar frequency, poor plaque control and xerostomia can increase the risk of developing caries [4]. Therefore, elderly patients [5,6] and specific risk groups such as patients after radiotherapy [7] are a particular challenge for caries management. This underlines the high topicality and demand for additive therapeutics, especially for biological approaches in preventive dentistry.

### 1.1. Role of the Pellicle at the Tooth Surface

When considering biofilm formation and demineralization processes, the pellicle, as the interface between the tooth surface and the oral cavity, plays an important role. It is a bacteria-free layer, which consists mainly of absorbed proteins, glycoproteins and other macromolecules [8]. The pellicle acts as a lubricant [8,9,10,11], plays a role in processes of de- and remineralization [8,11,12,13,14] and contains components with antibacterial properties [8,11,15]. In addition to these tooth-protective properties, the pellicle contains proteins that act as receptors for bacterial adhesion [8,16]. Therefore, it also forms the basis for bacterial adhesion and further biofilm formation. Fluorides [17,18,19,20], but also herbal agents such as polyphenols [21,22,23,24,25], lead to a modification of the pellicle layers and can enhance caries-preventive and erosion-protective properties.

### 1.2. Olive Oil and Its Perspectives in Preventive Dentistry

Lipids have key biological functions, serve as structural components and energy providers, and play a central role in the transduction of signaling pathways [26]. They are able to transport other macromolecules and substances, such as proteins and peptides, and are used in the transport of drugs [27]. Olive oil has anti-inflammatory and antibacterial properties, mainly due to its fatty acids oleic acid and linoleic acid and its polyphenol compounds, such as hydroxytyrosol, tyrosol, oleuopein and oleocanthal [28,29,30,31,32]. Since lipids have hydrophobic properties, it has been assumed that rinses with oil may also reduce bacterial adherence to the tooth surface [33,34]. In situ studies could neither demonstrate hydrophobization of the tooth surface nor anti-adherent effects of edible oils [35,36]. However, an incorporation of lipid micelles into the pellicle was shown on an ultrastructural level by transmission electron microscopy (TEM) and environmental scanning electron microscope (ESEM) techniques [35,37]. The present study aimed to investigate whether olive oil leads to an enhanced transport of bioactive molecules to the tooth surface.

### 1.3. Olive Oil as a Transport Medium of Bioactive Plant Compounds

Polyphenols are a group of natural compounds that provide biological benefits to plants [38]. Studies have shown that they also exhibit antibacterial, anti-inflammatory, antiviral and anticarcinogenic effects in the human organism [39,40,41,42]. They can denature pellicle proteins and strengthen the anti-adherent and antibacterial properties of the pellicle [21,22,25,43,44,45,46]. Rinsing with polyphenols leads to a thickening and strengthening of the pellicle layers, which enhances erosion-protective properties. In the present study, the lipophilic polyphenol ellagic acid (EA) was tested. Ellagitannins and EA are found in various fruits, seeds [47], roots [48] and nuts [49]. Particularly high levels of EA are found in strawberries, raspberries and blackberries [50,51,52]. Anti-inflammatory [53,54], antioxidant [55], antiviral [56,57] and antifungal [58] properties are attributed to the phenolic groups of EA. Rinsing with EA resulted in reduced growth of oral bacterial strains, reduced formation of water-insoluble glucans by *Streptococcus mutans* and reduced adhesion to the tooth surface [59]. Due to the ring compounds, it has a predominantly lipophilic character; the four phenolic groups and the two lactones form a hydrophilic zone (Figure 1) [60]. These lipophilic properties are relevant for the combination with oil.

The present study was designed to evaluate the reduction of bacterial adherence and the erosion-protective properties of EA in situ and to test whether the combination with olive oil leads to an enhancement of these properties. In situ studies are well suited for investigation of the pellicle. They take into account the dynamic changes of the oral cavity and allow the collection of pellicle material. Initial bacterial adhesion (after 8 h of wear) and glucan formation were evaluated. For the evaluation of the demineralization processes, calcium and phosphate release was determined photometrically. The study examines the hypothesis that olive oil acts as a carrier and can enhance the reduction of bacterial adherence and erosion-protective properties of the lipophilic polyphenol EA in situ.

## 2. Results

### 2.1. Initial Bacterial Adhesion

#### 2.1.1. DAPI/Glucan

The number of adherent bacteria was significantly reduced by rinsing with EA in water or in oil (Figure 2). Single bacterial cells and minor aggregates of bacteria were visible after rinsing (Figure 3). The number of adherent bacteria after rinsing with EA in water or EA in oil showed no significant differences. The glucan structures were significantly reduced after rinsing with EA in water (Figure 3).

#### 2.1.2. BacLight

Both rinses (EA in water/in oil) resulted in a significant reduction of vital and dead bacteria (Figure 3 and Figure 4). No significant differences were found between the two rinses, neither for vital nor for dead bacteria. Furthermore, no reduction in the ratio between vital and dead bacteria could be detected.

### 2.2. Calcium and Phosphate Release

Both rinsing solutions yielded a significant reduction in calcium release in contrast to both control groups (native, 30 min pellicle) at a pH of 3.0. After rinsing with EA in olive oil, significantly less calcium was released than after rinsing with EA in water (pH = 3.0).

For phosphate release (the 30 min pellicle), EA in water and EA in oil showed a significant reduction in contrast to the native enamel slab. At lower pH values (2.0 and 2.3), no significant differences were observed for either calcium or phosphate release (Figure 5 and Figure 6).

### 2.3. Transmission Electron Microscopy

The TEM images visualized distinct differences of the pellicle ultrastructure after rinsing with EA in water and EA in olive oil after an oral exposure time of 2 h. Whereas rinsing with EA in water led to a densification and thickening of the pellicle structure, rinsing with EA in olive oil resulted in the formation of a thin basal layer and loosened protein structures above (Figure 7).

## 3. Discussion

Various in situ studies have shown that polyphenols can reduce bacterial adherence and glucan formation at the tooth surface [21,22,24,25,45]. Thereby, whole extracts, such as *Cistus incanus* tea, seem to have stronger antibacterial and anti-adherent effects than their single fractions [45]. Besides this, there are also chemically manufactured polyphenolic single substances, such as epigallocatechin gallate or tannic acid, which show strong antibacterial and anti-adherent properties. For example, studies with tannic acid showed long-term antibacterial and anti-adherent effects after 24 h [46] and even 48 h [44].

In contrast to other polyphenolic whole-plant extracts, ellagic acid (EA) is a lipophilic polyphenol and is an industrially manufactured chemically derived substance like tannic acid. The aim of the present study was to evaluate whether EA in water and in oil can enhance the protective properties of the salivary pellicle in terms of anti-adherent and anti-erosive effects. The results show that rinsing with EA can lead to a significant reduction of vital and avital bacteria. However, in comparison to whole extracts such as extracts of fragaria vesca leaves, which are rich in EA, EA as a single compound shows less anti-adherent effects than the whole extract [22]. Consequently, whole extracts, with their interaction and synergy of polyphenolic compounds, seem to have stronger effects in terms of anti-adherent and anti-erosive properties.

For centuries, traditional medicine has used mixtures of plant extracts or whole-plant extracts instead of single substances or isolated plant compounds [61]. It seems that whole-plant extracts have greater in situ anti-adherent and anti-erosive activities than the single isolated components, as shown in the present study compared to a study with the same study design using whole-plant extracts of fragaria vesca containing high doses of EA [22]. Pure substances such as EA are industrially manufactured. They are produced or isolated from plants, in the hope that they will exhibit higher activity. However, as shown for anti-malaria activity, they did not present the same level of activity as the whole extract, even if the same concentration of active components was used [61,62]. Thereby, bioavailability and metabolism of a plant’s components are often affected by other plant components; these interactions are missing when using single substances such as EbA. The plant produces these chemical ingredients as defense mechanisms against natural enemies such as micro- and macroorganisms [61]. The high variety of chemical plant components covers a wide field of targets such as receptor sites and enzymes [61]; thereby, plants exhibit a complex amount and variety of phytochemicals such as polyphenols [61,63]. So far, little is known about these detailed mechanisms, and studies concerning pharmacokinetics or synergistic effects are necessary to gain more knowledge regarding active components of plants and their effectiveness.

In addition, pure chemically derived substances such as EA are often more cost-intensive than whole-plant extracts and are therefore unavailable for the general population [61,64]. EA is a lipophilic polyphenol. Avachat and Patel [65] demonstrated that it is suitable for transport by lipids. However, rinses with EA and olive oil showed no enhanced antibacterial properties. Previous investigations showed that edible oils do not have antibacterial and anti-adherent properties at the tooth surface [35]. After rinsing with silicon oil, oil vesicles with vital bacteria were observed in pellicle layers without alteration of the surrounding pellicle structure [66]. This finding indicates a preference of oral bacteria for hydrophobic surroundings [66]. The study of Hannig et al. supports this hypothesis, since there was a significantly higher number of adherent bacteria after rinsing with linseed oil. Rinses with olive oil and safflower oil led to equal amounts of bacteria, which was similar to the results of control groups with no rinsing [35]. The present study also shows that the combination of EA with olive oil does not lead to a reduction of adhered bacteria. Consequently, the combination of oil rinsing with a lipophilic polyphenol could not enhance the effect of the single use of polyphenols. As a result, olive oil seems unsuitable to serve as a carrier for EA.

A closer look at ultrastructural images (TEM) shows possible explanations. Rinsing with oil leads to a loosened pellicle ultrastructure [36]. In contrast, rinsing with polyphenols strengthens the pellicle ultrastructure through higher electron density and thicker pellicle layers [22,23]. The present study confirms these results. Rinsing with EA in water resulted in a higher electron density of the pellicle structure. In contrast, the combination of EA with olive oil yielded a thin basal layer and a loosened pellicle ultrastructure, which was rather thin compared with 120 min control samples as observed in previous studies [36,67]. Clearly, this combination does not optimize the ultrastructure of the pellicle in the desired manner. This is in line with previous studies on the effect of edible oils on the pellicle [36,68,69].

Several studies revealed that these effects—thickening of the pellicle and higher pellicle density—result in enhanced pellicle protection against bacterial adhesion and erosive attacks [22,23,25,46,70]. Conclusively, the positive effects of the polyphenol EA in combination with olive oil are not strong enough to compensate for the loosening of the pellicle ultrastructure produced by the oil. Polyphenols already interact with the protein structures of saliva by molecular forces, hydrophobic interactions and hydrogen bonding in the oral cavity [22,70]. Thereby, protein–polyphenol complexes are formed. Later on, these complexes adhere to the pellicle layer via crosslinking reactions, leading to a resistant and hard-to-remove pellicle layer on the tooth surface [71,72]. Typical pellicle and salivary proteins for polyphenol–protein interactions are histatins and prolin-rich proteins (PrPs). It was shown that polyphenols change the pellicle proteome [73]. Thereby, acid-resistant proteins such as statherin are increased, and the pellicle layers are harder to remove, leading to an anti-erosive effect [73]. A polyphenol-modified pellicle structure also has the ability to inhibit specific bacterial-derived and proteolytic enzymes, such as GTF, matrix metalloproteinase and α-amylase [74,75]. Through this inhibition, bacterial adhesion is disturbed, since glucan formation is reduced and receptor sites are blocked [21,24,45,74,76]. In addition, rinsing with polyphenols leads to lysis of bacterial cells [46].

Possibly, the hydrophobicity that results through rinsing with different oils prevents these protective protein–polyphenol interactions at the pellicle surface resulting in weaker protection against bacterial adherence and erosive attacks. The lipophilicity of the oil rinsing might also inhibit the attachment and interaction of polyphenol–protein complexes. An in vitro model showed that EA, as a chemical-derived substance, can reduce glucan formation [59]. This was also shown for the in situ investigation with EA in water in the present study. However, the combination with olive oil resulted in an increased glucan score than rinsing with EA in water. Previous results also failed to show a reduction in glucan structures after rinsing with edible oils [35]. Olive oil can be integrated into a polysaccharide matrix [77]. In addition, Lei et al. [77] have shown that lipid–water emulsions can be stabilized by polyphenols and polysaccharides. Thus, it is conceivable that the olive oil–EA–water emulsion is integrated into the glucan structure, and stabilized, and therefore leads to an increase in glucans. The combination with olive oil is therefore not recommended for the prevention of caries.

Like other rinsing solutions containing polyphenols, EA also could reduce demineralization processes at the tooth surface. Polyphenols lead to a thickening and strengthening of the pellicle layer. They can serve as protection against ion losses [23,25,78]. A particularly strong protection against demineralization was shown for a combined rinsing with whole extracts of Origanum and leaves of the Ribes nigrum [23]. Presumably, the diversity of different polyphenols improved erosion-protective properties. They led to a significant reduction of calcium and phosphate release at pH values 2.0, 2.3 and 3.0.

At the same time, there are polyphenols that have not shown any inhibition of demineralization, such as Inula viscosa tea [21]. The present study also shows that the erosion-protective properties of the modified pellicle by EA are limited. Only calcium release at pH 3.0 was reduced significantly in contrast to the physiological pellicle. No significant differences were shown at pH 2.0 and 2.3 or in phosphate release.

While the combination of EA with olive oil could not show any improvement in the reduction of bacterial adherence, the combination EA with olive oil led to enhanced erosion-protective properties.

It is likely that the incorporation of lipid micelles into the pellicle, which is evident after oil rinses [35], leads to the formation of a diffusion barrier. Studies show that lipid structures can enhance the erosion-protective effect of fluorides [79,80]. In addition, after rinsing with Origanum and leaves of the Ribes nigrum lipophilic, components could be visualized in TEM images [23]. Thus, the present study confirms the assumption that olive oil accumulates on the tooth surface and can act as a diffusion barrier.

Nevertheless, it must be mentioned again that the protection in combination with olive oil was also limited. Significant differences were only shown for calcium release at a pH of 3.0.

## 4. Material and Methods

### 4.1. Subjects

Six volunteers (25–40 years) participated in the present study. Prior to the study, visual oral examination was performed by an experienced dentist. The participants showed no signs of caries, gingivitis, periodontitis or other diseases of the oral cavity. All subjects were non-smokers and had no signs of dental erosion. The study design was reviewed by the ethics committee of the University of Dresden (EK 147052013).

### 4.2. Specimens

The test specimens were obtained from permanent bovine incisors (BSE-negative animals). The adoption of this animal by-product was approved by the responsible governmental agency: EC 147,052,013 (Landesdirektion Sachsen, Germany). Enamel specimens were prepared from the labial surfaces: a diameter of 5 mm for fluorescence microscopy and calcium and phosphate release; and 2 × 2 mm for transmission electron microscopy (TEM). First, the dentin side was ground flat with wet grinding abrasive paper (320 grit). Afterwards, the enamel was polished (400–4000 grit) [81]. The specimens for erosion testing were first sealed on the dentin side using Optibond FL (Prime & Adhesive; Kerr dental, Bioggio, Switzerland) before grinding the enamel side. For this purpose, the test specimens were etched on all sites except for the outer enamel surface with 37% phosphoric acid (ScotchbondTM Universal Etchtant; ESPE, Neuss, Germany) for 30 s. Primer was applied with brushing motions before adhesive was applied three times and light-cured in a halogen light furnace for 30 s each time.

To remove the smear layer, the specimens were first cleaned in 3% sodium hypochlorite for 3 min. Then they were rinsed with distilled water for 5 min twice and disinfected with 70% ethanol for 10 min using an ultrasonic bath (US), and finally washed in distilled water again. For the formation of the hydrate shell, the test specimens were stored in distilled water at 4 °C [36,82].

### 4.3. Preparation of EA in Water/in Oil

For the preparation of EA in water, 2 mg of EA was added to 8 mL of distilled water. A deprotonation of the acid group was achieved by adding 12 µL of 1 M NaOH solution. The EA was dissolved in an ultrasonic bath at a temperature of 50 °C for 30 min. For the preparation of EA in oil, 2 mg of EA was dissolved in 4 mL of distilled water. Dissolution was also obtained by adding 12 µL of 1 M NaOH solution (US, 30 min, 50 °C). Both solutions were deep frozen at −20 °C until use.

The mix of EA–water solution with oil was prepared prior to the experiment. A solution of 2 mg of EA/4 mL was defrosted and mixed with 4 mL of olive oil on a shaker. The pH values for the solutions were 7.48 for EA in water and 7.56 for EA in oil.

### 4.4. Pellicle Formation

For the experiments, individual wire/polymethyl methacrylate splints were prepared [35,83]. The test specimens were fixed buccally at regions 14–16 and 24–26 by using polyvinyl siloxane impression material (Light regular set; Kulzer GmbH, Hanau, Germany). The subjects were instructed to brush their teeth without toothpaste and to refrain from eating and drinking except for water prior to the experiment. After insertion of the splint and a pellicle formation time of 1 min, the volunteers rinsed for 10 min with EA in water or EA in oil. For the control, the splints were worn without rinsing for the same time (Figure 8).

To investigate bacterial adherence, the splints were worn overnight for a total of 8 h and evaluated by fluorescence microscopy (DAPI, ConA, BacLight). For investigation of demineralization processes, the enamel slabs were acidified after a wearing time of 30 min ex vivo with HCl (pH = 2.0, 2.3 and 3.0), and calcium and phosphate release was determined for 120 s every 15 s (Figure 2). The modification of the pellicle ultrastructure was evaluated by transmission electron microscopy after an oral exposure time of 2 h.

### 4.5. Initial Bacterial Colonization and Glucan Formation

Specific fluorescent dyes were used to visualize adherent bacteria and glucan formation at the enamel surface. For the investigations, the test specimens were examined under a fluorescence microscope (Axioskop II; Zeiss, Oberkochen, Germany) at 1000-fold magnification. Five representative areas were selected to count the number of adherent bacteria in an area of 100 µm × 100 µm. This allowed the calculation of the number of bacteria per square centimeter.

#### 4.5.1. DAPI/Glucan

The staining procedure was conducted as described previously [81,83]. DAPI (4′,6-Diamidino-2-phenylindol; Merck, Darmstadt, Germany) is a fluorescent dye that binds to AT-rich sequences of DNA. This binding leads to an excitation of the DAPI molecule, resulting in strongly fluorescent nuclei. The lectin ConA (Concavalin A, Alexa Fluor 594 conjugate; Invitrogen Ltd., Carlsbad, CA, USA) was used to investigate the glucan structures. This binds selectively to glucans, and consequently they are marked by red fluorescence.

The worn test specimens were removed from the splint, washed in 0.9% NaCl and then incubated in a solution consisting of 245 µL of buffer solution, 5 µL of ConA stock solution and 0.75 µL of DAPI stock solution. After 15 min of staining in a dark chamber, the specimens were washed in 0.9% NaCl and air dried. Afterwards, they were microscopically examined with a DAPI light filter (BP 381-399, FT 416 and LP 430–490) and a Texas Red light filter (BP 542–576, FT 585 and LP 595–664). By overlaying the images obtained with both light filters, the Axio Vision program allowed simultaneous assessment of the number of adherent bacteria and glucan structures. Glucan formation was evaluated using the glucan score.Score 0: no glucans detectable;Score 1: single glucan structures exhibiting cloudy formations;Score 2: distinct glucan structures exhibiting cloudy structures surrounding the bacteria;Score 3: distinct glucan structures surrounding at least 50% of the bacteria;Score 4: distinct glucan structures surrounding almost all bacteria.


#### 4.5.2. BacLight

The LIVE/DEAD^®^ BacLightTM Bacterial Visibility Kit (Invitrogen, Molecular probes, Darmstadt, Germany) is used to distinguish between vital and avital bacteria. It is based on two nucleic acid stains. Syto^®^9 has an excitation and emission maximum of 480/500 nm and is green fluorescent. It penetrates bacterial cells with intact and defective membranes. Propidium iodide has an excitation and emission maximum of 490/635 nm and fluoresces red. It only penetrates cells with a destroyed bacterial cell membrane, but is able to displace Syto^®^9 from DNA. Therefore, a differentiation between green-fluorescing vital cells and red-fluorescing avital cells is possible.

After removal of the specimens from the splints and washing with 0.9% NaCl, they were incubated in a dark chamber with 0.5 μL each of components A and B in 500 μL of 0.9% NaCl for 10 min. The enamel samples were washed again and air dried prior to fluorescence microscopic analysis using a fluorescein diacetate light filter and an ethidium bromide light filter. The images were overlapped using the Zeiss-Axio-Vision program.

### 4.6. Calcium and Phosphate Release

After oral exposure (30 min), the enamel slabs were embedded in a 2 mL Eppendorf cup with Provil^®^ novo light regular (Kulzer GmbH, Hanau, Germany) to determine calcium and phosphate release from the upper enamel surface. The physiological 30 min pellicle without any rinsing and native enamel samples without oral exposure served as controls. The specimens were then acidified with different concentrations of hydrochloric acid for 2 min (pH = 2.0, 2.3 and 3.0). For this purpose, 1000 μL of hydrochloric acid was added to the sample. Every 15 s, 100 μL of the solution was removed for photometric analysis and replaced by 100 μL of fresh acid. From each of these test volumes, a triplicate determination of 10 μL was carried out for calcium release and likewise for phosphate release [23,25].

Arsenazo III binds to calcium ions and forms blue-color complexes that can be measured photometrically at 650 nm [84]. Binding of phosphate ions to the malachite green solution forms green-color complexes, which can also be measured at 650 nm [85].

The total release of one enamel slab was calculated from the measured data. Since the dissolution of the calcium and phosphate resins was almost linear, statistical evaluation was carried out with the cumulative values after 120 s.

### 4.7. Transmission Electron Microscopy

After fixation of the samples in glutaraldehyde (2.5% glutaraldehyde, 1.5% formaldehyde in phosphate buffer and pH 7.4) for 2 h, the samples were washed five times in a phosphate buffer. To visualize organic structures, they were incubated in 1% osmium tetraoxide for 2 h. The samples were dehydrated using ethanol with a rising gradient concentration and embedded in Araldite M (Serva, Darmstadt, Germany). Ultra-thin sections were cut in an ultramicrotome (Ultracut E; Reichert, Bensheim, Germany) with a diamond knife (Microstar 45°; Plano GmbH, Wetzlar, Germany). The specimens were mounted on Pioloform-coated copper grids and contrasted with uranyl acetate and lead citrate. They were examined in a TECNAI 12 Biotwin TEM with a magnification of 30,000–49,000, at which representative images were taken (50–100 images) [36,67].

### 4.8. Statistics

Statistical evaluation was carried out using Kruskal–Wallis tests, Mann–Whitney U tests (*p* < 0.05) and subsequent Bonferroni–Holm correction. The software used was SPSS statistics 27.0 (IBM, Ehningen, Germany).

## 5. Conclusions

The combination of olive oil with EA cannot improve antibacterial properties. Olive oil is not suitable as a transport medium for lipophilic polyphenols. However, the emulsion of EA in water and olive oil seems to lead to an accumulation on the tooth surface, which acts as a diffusion barrier and enhances erosion-protective properties. However, neither modification of the pellicle by EA in water nor in olive oil led to significant protection against ion losses at pH values 2.0 and 2.3.

Particular attention should be directed to the pH values of the rinsing solutions used in the present study. Both rinsing solutions have neutral pH values (EA in water, 7.48; EA in olive oil, 7.56) and could be an option for patients with xerostomia and mucositis. EA could therefore be a prospect for use in caries prophylaxis in radiotherapy patients.

## Figures and Tables

**Figure 1 molecules-28-03803-f001:**
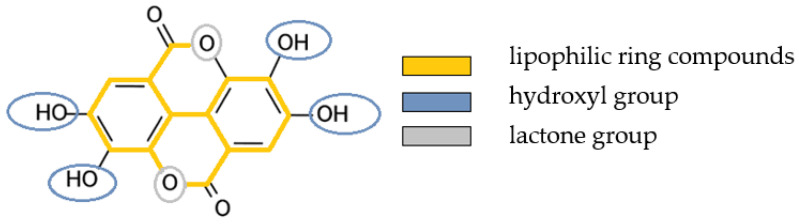
Structural formula of ellagic acid and its lipophilic and hydrophilic compounds.

**Figure 2 molecules-28-03803-f002:**
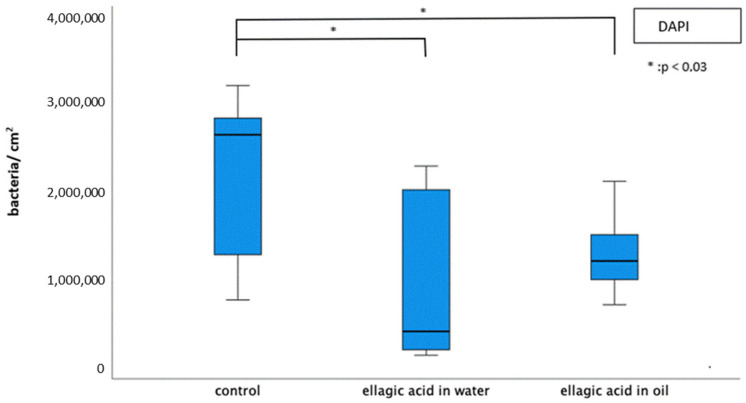
Boxplot diagram of DAPI fluorescence staining for the detection of adherent bacteria after 8 h biofilm formation with/without rinsing with EA in water or in oil. EA in both water and in oil led to a significant reduction in the total number of bacteria (Kruskal–Wallis test (*p* < 0.008), Mann–Whitney U test (*p* < 0.05) and Bonferroni–Holm correction (*p* < 0.03)).

**Figure 3 molecules-28-03803-f003:**
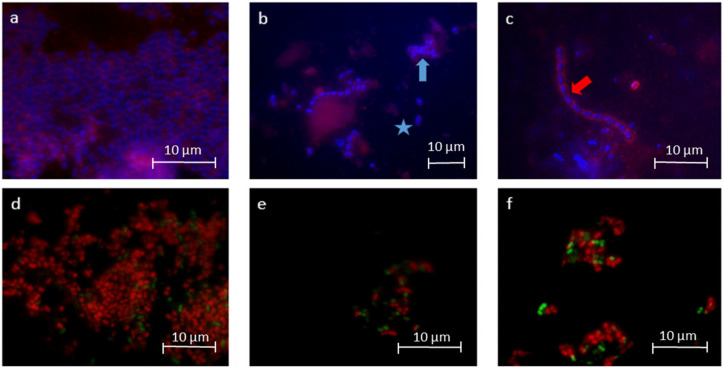
Representative fluorescence microscopic images after DAPI/ConA staining (**a**–**c**) and BacLightTM staining (**d**–**f**). The images visualize the reduction of the total number of bacteria (blue) after rinsing with EA in water (**b**) and in oil (**c**) in contrast to the control (**a**). The fluorescence microscopy images show single bacterial cells (blue asterisk, **b**) but also bacteria in small aggregates (blue arrow, **b**). The glucan structures (red, **a**–**c**) were reduced after rinsing with EA in water (**b**). Particularly after rinsing with EA in oil, the bacteria showed formations of chains (red arrow, **c**). BacLightTM staining was used to differentiate between vital (green) and avital bacteria (red, **d**–**f**). Both the number of vital and avital bacteria decreased after rinsing with EA in water (**e**) and in oil (**f**) in contrast to the control (**d**).

**Figure 4 molecules-28-03803-f004:**
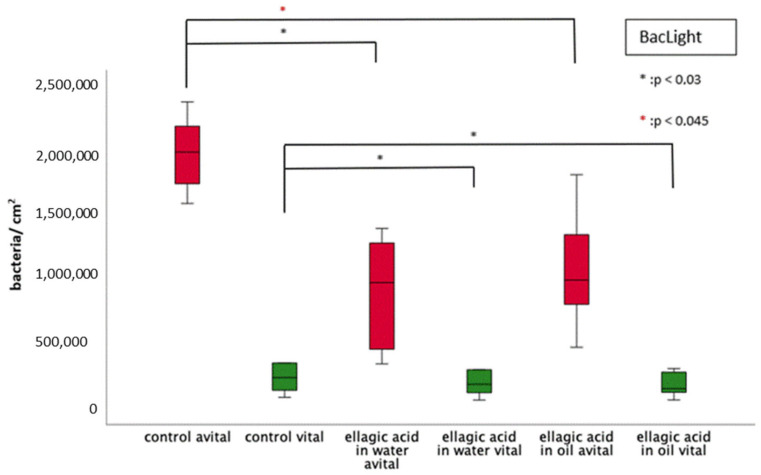
Boxplot diagram of the BacLight viability staining for the detection of vital/avital adherent bacteria/cm^2^ after 8 h of biofilm formation with/without rinsing with EA in water or oil. The number of avital bacteria is shown in red and number of vital bacteria in green. EA in water and in oil led to a significant reduction of avital and vital bacteria (Kruskal–Wallis test (*p* < 0.05), Mann–Whitney U test (*p* < 0.05) and Bonferroni–Holm correction (*p* < 0.045 and 0.03, respectively)). There was no significant difference between the rinsing of EA in water and in oil.

**Figure 5 molecules-28-03803-f005:**
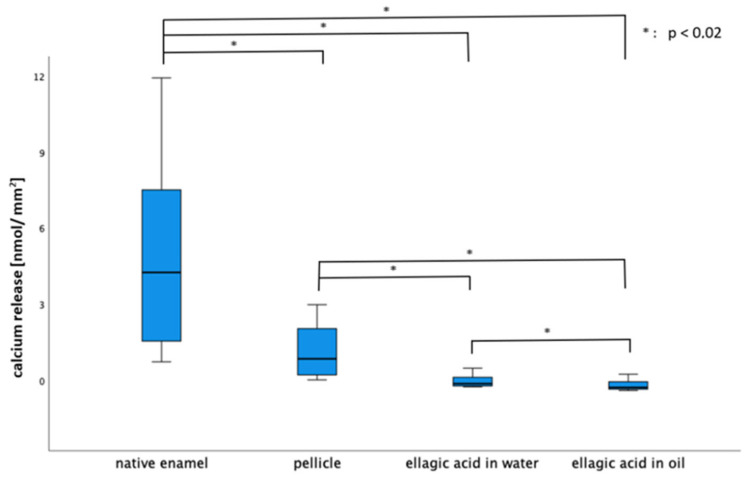
Boxplot diagram of calcium release of the native enamel, enamel with an in situ pellicle and with a pellicle after rinsing with EA in water and in oil (pH = 3.0). The pellicle led to a significant reduction in calcium release in comparison to the native enamel slab. The EA in water and in oil led to a significant reduction in contrast to both control groups (native enamel and pellicle). EA in oil showed significant less calcium loss than EA in water (Kruskal–Wallis test, Mann–Whitney U test and Bonferroni–Holm correction, *p* < 0.02).

**Figure 6 molecules-28-03803-f006:**
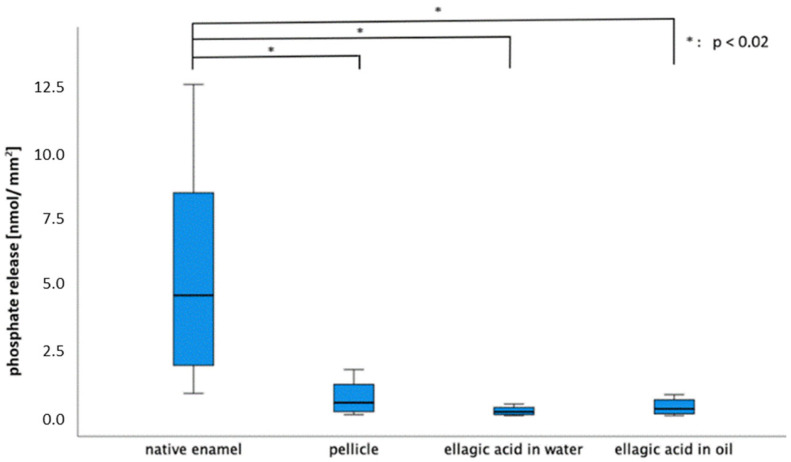
Boxplot diagram illustrating the phosphate release of native enamel, enamel with a 30 min pellicle and of the tooth surface after rinsing with EA in water and in oil (pH = 3.0). The phosphate release was significantly reduced after pellicle formation and after both rinses in contrast to the native enamel slab (Kruskal–Wallis test, Mann–Whitney U test and Bonferroni–Holm correction, *p* < 0.02).

**Figure 7 molecules-28-03803-f007:**
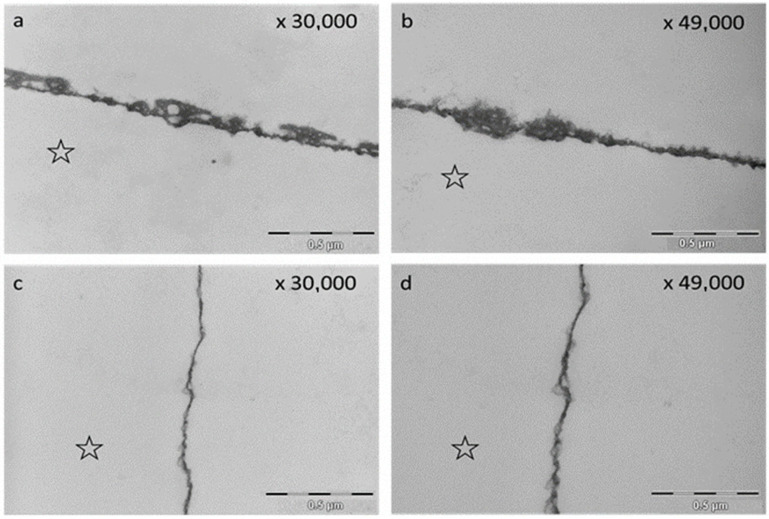
Representative TEM images of the in situ pellicle after rinsing with EA in water (**a**,**b**) and EA in olive oil (**c**,**d**) (oral exposure: 2 h). The rinsing with EA in water led to an increase in pellicle thickness and electron density (**a**,**b**). Image (**a**) shows an inclusion of vesicles and irregular vacuoles. The rinsing with EA in oil shows a thin basal pellicle layer and loosened protein structures above (**c**,**d**). The former enamel site is marked with an asterisk.

**Figure 8 molecules-28-03803-f008:**
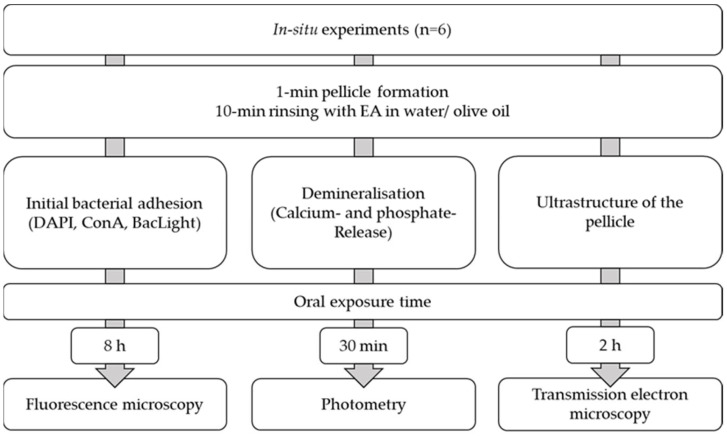
Flowchart of the in situ experiments.

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
