# Peer review of "Olive Oil as a Transport Medium for Bioactive Molecules of Plants?—An In Situ Study"

_molecules, 2023, doi:10.3390/molecules28093803_

Round 1

Reviewer 1 Report

A brief explanation of each section of the report is given below;

Title: 
It is adequate for the content, informative, concise, and clear.

 Abstract: It is comprehensive by itself. All the important and essential information of the article is included. Structure and length: The overall structure of the article well organized and well balanced. The article is written with the minimum length necessary for all relevant information.
 Logic: The article written clearly and correctly. It is logically consistent. Figures and tables: They essential and clearly presented. English: English used in the article should be improved. There are some typographic and grammatical errors. Scientific quality rating: The article is novel and original. The article contains material that is new or adds significantly to knowledge already published. Importance and impact: The presented results are of significant importance and impact to advancement in the relevant field of research. References: Appropriate and adequate references to related works covered sufficiently in the list. Incorporate some latest references.

Minor Revision :

Problem, Need, scope of the project is missing. Figure 3 images at 10 um are not much clear. Pellicle formation section should be improved. The article should become acceptable after minor revisions of English / grammatical mistakes / improvement of figures quality.

Author Response

Dear Editor,

Please find enclosed our revised manuscript. We would like to thank the reviewers for their helpful comments, the issues were answered one by one as given below; changes are marked in yellow in the main document

Kindest regards in the name of the authors

C Meyer-Probst

Reviewer 2 Report

Title: Clearly reassumes the content of the manuscript.

Abstract: Although the abstract extensively describes methods and results it misses the rational and the impact of this research, as well as author’s conclusion  

Introduction: It is overall well written with the background and the aim.

Material and method sections: appropriate

Results: As also reported below minor revision is required

Discussion and Conclusion: are overall clear and linear

 1. The authors attempt to evaluate the role of EA as antibacterial as well as its anti-adherent and erosion-protective properties, employing complementary methods

2. EA and olive oil (individually or in combination) antimicrobial functions have been extensively studied in several different contexts. However, this article represents a novelty in this particular field and gives new inputs to the knowledge, as the authors find partially enhanced erosion-protective properties of EA emulsion in olive oil, although no improvement as antibacterial properties was detected.

3. The presented results might have beneficial impact in patients with xerostomia and mucositis since the authors find the optimal working pH (neutral in this particular case) for EA emulsion, which is significantly relevant in the field and adds new medical options.

4. The reported methodology approach and controls are appropriate

5. The conclusions are consistent and address the scientific questions underlying the logic of the presented paper.

6. All references are properly listed and appropriated for the presented context

Beside some grammatical changes in the result/discussion sections, I point out that there are two Fig.3, while Fig. 4 is missing, and the scale bar are different among the showed panels; quality should be improved if possible. The fluorescence images are representative of “are out of many?”. 

Fig 8, scale bar is missing. TEM images are representative of “are out of many?”. 

Author Response

(The authors gave the same response as above.)
